# Employee engagement, perceived organizational support, and job performance of medical staff at the Cape Coast Teaching Hospital

**Felix Kwame Opoku**, **Richard Kofi Boateng***

Department of Human Resource Management, School of Business, University of Cape Coast, Cape Coast, Ghana

* kofirichboateng1994@gmail.com

## Abstract

This study examined the moderating effect of perceived organizational support on the link between employee engagement and the job performance of medical staff at the Cape Coast Teaching Hospital. The study adopted a purely quantitative approach, using the explanatory research design. Data were collected from 310 full-time nurses and midwives of the Cape Coast Teaching Hospital in the Central Region of Ghana. The available data were analyzed using the Structural Equation Modelling technique. The study's results revealed that employee engagement significantly influences the job performance of nurses and midwives in the Cape Coast Teaching Hospital. The study further revealed that perceived organizational support moderates the link between employee engagement and the job performance of nurses and midwives in the hospital. Given these findings, the study recommended that to effectively improve the job performance of nurses and midwives, the management of the Cape Coast Teaching Hospital must adopt policies such as effective leadership and fair HR practices to stimulate employee engagement and their perceived organizational support.

## Introduction

Employee engagement is lately emerging as one of the vocabularies of prime concern within the functional unit of Human Resource Management [1]. Recent statistics show that improved employee engagement in healthcare correlates with higher patient satisfaction and reduced medical errors. For example, a study by [2] reported that engaged healthcare employees were 25% more likely to exceed patient care expectations, leading to better overall health outcomes. Similarly, [3] found that hospitals with higher employee engagement scores reported a 15% increase in patient satisfaction and a 20% decrease in medical errors. Indeed, previous studies [4–6] have found that employee engagement is a strong predictor of several positive organizational outcomes. For instance, [4] conducted a study on the effect of employee engagement on job performance and found that employee engagement is a strong positive predictor of job performance. Again, [5] found that employee engagement increases employees' organizational

**Competing interests:** NO. The authors have declared that no competing interests exist.

commitment which in turn improves their job performance. [6] also conducted a study on the impact of employee engagement on work satisfaction and job performance among mining workers in Indonesia. The authors found that employee engagement is a positive and significant predictor of both worker satisfaction and job performance. Some empirical studies have also found that employee engagement reduces employee turnover intentions [7] and other counterproductive employee behaviours [8, 9].

Although a plethora of studies have examined and reported the significant and positive link between employee engagement and job performance, there are only two studies in the literature that attempted to investigate personality and situational factors, moderating this link. [10] For instance, examined the moderating role of a personality factor (conscientiousness) between work engagement (job characteristics) and job performance. The results of that study supported the hypothesis that work flow predicted in-role and extra-role performance for only conscientious employees. [11] also investigated the moderating effect of a situational factor (perceived organizational support) on the link between work engagement and job performance in South China. The authors found that the positive relationship between employee engagement and job performance is more significant when perceived organizational support is higher than lower. However, whereas these two studies can be mentioned about developed countries, there is no single study examining the moderating role of either personality or situational factor between employee engagement and job performance in a developing country context.

The current study fills this gap by investigating the moderating effect of perceived organizational support on the link between employee engagement and job performance of nurses and midwives in the Cape Coast Teaching Hospital. Though a similar study has been conducted by [11], their study was conducted in China where employees' backgrounds, cultures, and beliefs of employee engagement, organizational support and job performance are different from Ghana. Ghana as a developing country lacks the advanced equipment to support health workers as compared to China. Also, the study by [11] only considered one dimension of job performance (task). However, other dimensions (contextual and adaptive) of job performance can equally be affected by the level of employee engagement and support given to workers by the organization. Furthermore, their study used customers in the telecommunication industry as a unit of analysis and taking recommendation from this study may be misleading as the current study seeks to examine employees in the health sector which has unique contextual factors [12]. Again, as used in [11], work engagement only relates to the relationship of employees with their work. However, the current study used employee engagement to include the relationship with other employees and the organization as a whole. Thus, despite previous research on the moderation of perceived organizational support on the link between employee engagement and job performance among employees in the Chinese telecommunication industry, there is a research gap in understanding this relationship among employees in other industries and regions.

By conducting this study, we intend to address critical gaps in the current literature on employee engagement and job performance within the context of developing countries, specifically Ghana. Despite the substantial evidence linking employee engagement to positive organizational outcomes, studies investigating the moderating role of perceived organizational support are predominantly centered in developed countries. This creates a pressing need to explore how these dynamics unfold in developing nations with unique cultural, economic, and infrastructural challenges. The healthcare sector, particularly in Ghana, faces significant constraints, including limited resources and differing employee expectations, which can impact on job performance and engagement. By focusing on nurses and midwives at the Cape Coast

Teaching Hospital, this research aims to provide insights that could enhance employee engagement strategies, improve job performance, and ultimately lead to better healthcare delivery.

Additionally, understanding these relationships in a developing country context can inform policy and management practices, making the findings highly relevant for both academic and practical applications. The findings of this research would help hospital management and policy-making in the healthcare sector understand the role of perceived organizational support in enhancing employee engagement. This would help the hospital management to implement strategies that foster a supportive work environment. This can lead to higher job performance, reduced turnover rates, and improved patient care. For instance, policies focusing on effective leadership, fair HR practices, and career development opportunities can enhance employee engagement and performance. Additionally, providing adequate resources and support can address the unique challenges faced by healthcare workers in developing countries like Ghana, ultimately leading to better healthcare outcomes and more efficient hospital operations.

## Literature review and development of hypotheses

This section examined the theoretical and conceptual issues underlying the study. The section is organized into two: (1) theoretical review, and (2) conceptual review.

### Theoretical review

**Social exchange theory.** This study is underpinned by the social exchange theory, proposed by [13]. In proposing this theory, [13] drew from the concept of behaviorism by comparing human behaviour to the behaviour of pigeons whenever they receive corn for pecking a target. He also drew from his own works on social influence, equilibrium, cohesiveness, and conformity as they relate to small groups [14]. The theory postulates that social behaviour is the result of an exchange process, whose purpose is to maximize benefits and minimize costs. According to [13], an individual will evaluate the cost of a social interaction (bad consequence) against the reward of that interaction (positive outcome). The social exchange theory thrives on four constituents of social behaviour. First, the theory defines reinforcement tools–that is, the rewards-cost and resources of exchange that underlie the individuals' motivation to engage in social interaction. These costs and rewards can be material such as, money, time, or a service [15]. They can also be intangible such as effort, social recognition, love, pride, humiliation, respect, opportunity, and power [15].

The second constituent (mechanisms of exchange) postulates that people are generally rational and would always weigh the costs and savings of social interactions analysis [14]. In this regard, individuals exist as rational actors as well as reactors in social exchanges. Thus, the second constituent is contingent on two conditions that define the decision of the person to engage in exchange relations. The third constituent of the social exchange theory is reciprocity, which creates obligations among the actors in social exchange relations [16]. In this regard, the actors in social exchange processes are rationally attempting to maximize the profits or benefits to be derived from those interactions, particularly in terms of addressing basic individual needs. The social exchange processes are stimulated by social structures and social capital factors [14]. The third constituent is possible because humans are evolutionarily predisposed to behave in such a way as to ensure reciprocation. Finally, the theory holds that exchange mechanisms that result in pay-offs or rewards for the individual forms a social interaction patterning. These patterns of social interaction do not only satisfy individual's wants, but also inhibit individuals' ability to meet those needs in the future.

The social exchange theory helps members of a society or social group to understand that because of the competitive nature of social systems, exchange processes would consistently

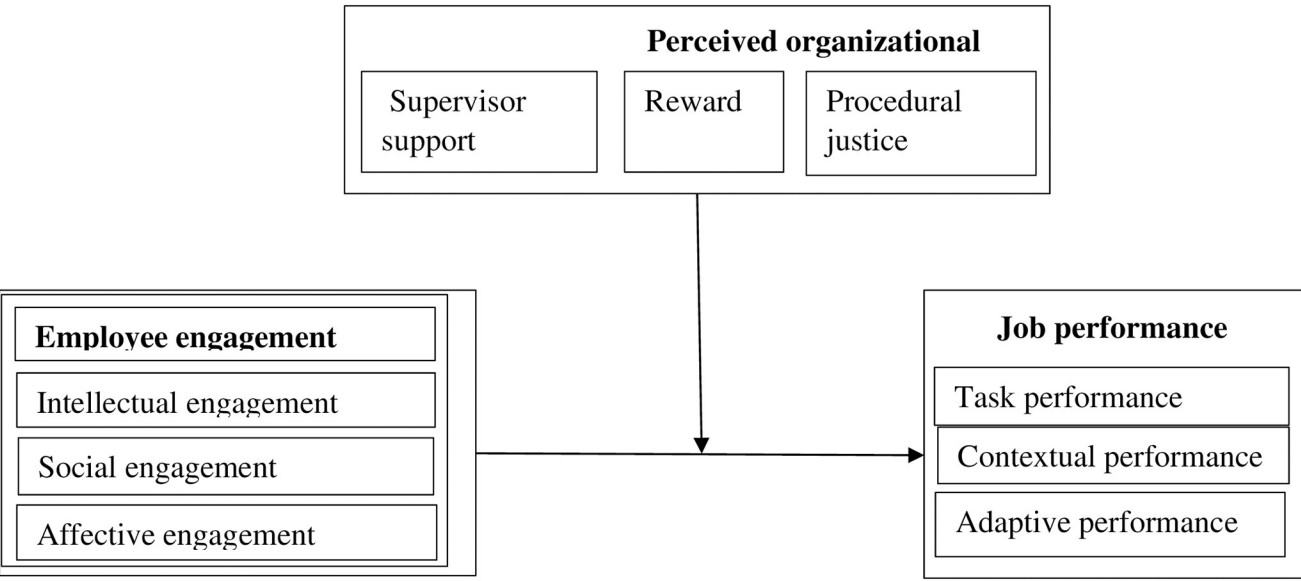

**Fig 1. Conceptual framework for the study.**

result in power and privilege difference within social groupings [17]. In this wise, those with greater resources exert more authority and, as a result, stand to benefit more from the interactions. Thus, the theory sees society as a location where individuals and groups can trade resources [18]. This is based on the notion that a person may continue in the long-run if he perceives that reciprocity will not be violated by other members of the group (12). In this way, the social exchange theory is built on the premise that people form social ties in order to maximize their collective ability for leveraging resources and mobilizing collective action in their respective environments (15). The relationships between the variables are presented in Fig 1.

## Conceptual issues

**Concept of employee engagement.**   [19] defined employee engagement as a state in which individuals feel dedicated, involved, empowered, and enthusiastic about their work, and express these sentiments in their work behaviour to attain better performance. The concept is also defined by [20] as a positive work-related state of the mind that sparks employee belief in three dimensions: intellectual engagement, social engagement, and affective engagement. Intellectual engagement portrays an employee's understanding of their tasks in connection with organizational goals and objectives. According to [21], an employee is intellectually engaged when he is cognitively awake and contextually aware of their tasks as well as the broader organizational aims and objectives. Intellectual engagement manifests itself in the form of employee dedication, awareness, and heightened attention to organizational goals and objectives [22, 23]. The degree to which individuals feel socially connected in their work environment and share similar values is termed social engagement [20, 22]. Social engagement allows employees to interact with one another on a social level and establish ties beyond professional relationships. It provides employees with a sense of belongness to a group and of being a part of something bigger than themselves. According to [21], the socially engaged employee is an individual who can adapt, initiate, and respond to changing organizational situations. The third and final dimension—affective engagement—depicts employees' attitudes

toward the organization, its leaders, and working circumstances, as well as their emotional bonds with coworkers and superiors [21]. It is typically seen in terms of trust and perceived organizational support. It is conceptually related to organizational commitment, work participation, and job satisfaction [17].

Three levels of employee engagement are frequently recognized by both academics and practitioners in the literature–actively engaged, not engaged and disengaged [23]. Actively engaged employees are often described as "alpha players" of the organization. They are the group of employees who perform their tasks and duties with passion, enthusiasm and pleasure. According to [24], the actively engaged employee is naturally curious about all activities in the organization and would contribute in the most productive and innovative ways to ensure that the organization achieves her goals and objectives. Actively engaged employees bring fresh ideas and ingenuity during problem-solving and thrive in collaborative contexts with coworkers [25]. Not engaged employees on the other hand, just concentrate on duties assigned to them by their supervisors. They simply report to work, perform the duties in their job description and would not go beyond. Finally, disengaged employees are the group of workers who are present and absent at the same time. They attend work every day but do not put much effort in their work. They are clock-punchers whose contribution to performance is minimal. Disengaged employees try to resist the feeling of affiliation to the organization as much as possible, and views the organization only as a check payer every month. They are reluctant to devote time to team-building activities or professional development opportunities that might help them advance in their careers at the organization [25]. They are not only unhappy at work but can also undermine the efforts of their engaged co-workers.

## Concept of job performance

[26] describe job performance as the overall effectiveness and achievement of an individual over a given period in fulfilling job responsibilities relative to set standards, goals, or agreed-upon criteria. Similarly, [27] characterize job performance as the measurable activities, behaviors, and results of employees that are associated with and support the attainment of organizational objectives. Borman and Motowidlo's (1993) classified job performance into task and contextual performance. Task performance refers to the set of employee behaviours that are directed at the job itself, and intended to accomplish the job requirements as provided in the job description [28]. They are the job explicit behaviours that are applied to do the job. According to [29], task performance refers to the more technical aspects of job performance that requires the application of skills, knowledge, and cognitive abilities. In the context of the health sector, task performance may include successful birth, reduced neonatal and maternal death, improved utilization of out-patients and antenatal services, time spent at the OPD and provision of holistic care at all service points.

[30] defined contextual job performance to include behaviours that are directed towards an individual, group or organization by a member while carrying out his assigned task. They are the variety of non-specific-job behaviours which indirectly support the efficient accomplishment of the entrusted task. According to [31], contextual job performance behaviours include supporting organizational decisions for a better change, helping others to solve difficult problems, upholding enthusiasm at work, cooperating with others in time of need, and volunteering for extra work. Thus, contextual behaviours go beyond the employees' task performance and technical proficiency. Contextual attitudes often create a great work environment for everybody [32]. [33] identified a third dimension of job performance known as the adaptive job performance. Adaptive performance refers to the ability of employees to adjust to and comprehend job changes and changing work settings [32]. Employees with adaptive job

performance are versatile and can solve problems imaginatively, dealing with uncertainty, learning new tasks, exhibiting interpersonal flexibility, and dealing with crises [32]. Employees who exhibit strong adaptive performance have a competitive edge in terms of career opportunities.

## Concept of perceived organizational support

[34] defined perceived organizational support as employee's belief about the extent to which the organization values their contributions, and cares about their well-being. According to [35], employees with strong perceived organizational support are more dedicated to the organization and are more satisfied with their jobs. According to [36], perceived organizational support is a one-dimensional criterion, covering employees' common belief that their organization is committed to their needs or continued membership in the organization. According to [37], organizational support goes through three processes that are beneficial to both employees and the organization as follows:

In accordance with the principles of reciprocity norms, it has been observed that organisational support has the potential to cultivate a sense of responsibility among employees, prompting them to consistently prioritise the well-being of the organisation and demonstrate a willingness to contribute towards the attainment of the organization's objectives. One possible explanation for this phenomenon is that employees perceive their investment in the organisation as highly valuable and feel a sense of responsibility towards it. Additionally, the provision of organisational support can effectively meet the social and emotional needs of employees, thereby increasing the likelihood of their continued membership within the organisation. The fulfilment of social and emotional needs within the workplace is facilitated by the organization's consistent demonstration of care and provision of fairness and appreciation to its employees. Furthermore, the provision of organisational support serves to enhance employees' trust in the organization's ability to reward them in accordance with their increased productivity, thereby reinforcing the performance-reward expectancies. The aforementioned processes exhibit advantageous characteristics for both the organisation and its employees.

[38] also identified three managerial roles for improving perceived organizational support among employees as follows:

1. building a feeling of approval which is usually based on the overall quality of superiors' behaviour towards employees, especially trust, which can be more important than an action or any combination of actions;

2. developing personal relationship to get to know the subordinates and also to help solving their problems inside, and outside, work;

3. providing a fair treatment by letting every employee knows what is expected of them and by putting discipline in place.

To achieve the preceding roles, HR managers must adopt favourable HR practices, effective leadership style, fair treatment and desirable job conditions [35]. Thus, any type of award that comes from the organization and demonstrates that the organization is satisfied and pleased with an employee's work, contributes to that employee's perceived organizational support. From an employee's perspective, the relationship between them and their supervisor is perhaps the single most important determinant of perceived organizational support [39]. As a result, because supervisors are representatives of the company, interactions between a supervisor and an employee might impact on the employee's perception of

organizational support. According to [40], any fair practices used by the organization demonstrate to workers that the organization appreciates them. Organizations that take steps to promote procedural fairness show their employees that they are valued and supported. As a result, employees' perceptions of procedural justice impact their perceptions of organizational support.

**Employee engagement and job performance.** A plethora of studies have examined the positive nexus between employee engagement and job performance [25, 41–45]. [41] investigated the impact of employee engagement on performance of employees in India and found that employee engagement had a significant positive impact on job performance. [25] also found that some drivers of employee engagement such as communication and leadership had a significant effect on job performance. In a study conducted by [45] to examine the effect of employee engagement and job performance among16 selected banks in Ethiopia, it was found that employee engagement has a significant and positive effect on job performance. Again, [43] investigated the effect of employee engagement and job performance in Lebanon and found that employee engagement has a significant, positive effect on employee's job performance. [44] conducted a study to determine the drivers of employee engagement and its effect on job performance in Pakistan. The authors found that employee engagement influences employee performance positively and significantly. Based on these reviews, it is hypothesized that:

*H₁*. *Employee engagement has significant effect on their job performance.*

**Perceived organizational support, employee engagement, and job performance.** In this study, it is argued that perceived organizational support moderates the positive link between employee engagement and job performance. This argument derives from the social exchange theory that was originally proposed by [13] and expanded upon by [14]. According to this theory, organizational support can create a sense of obligation which improves the employees' physical, social and cognitive attachment to the organization and its well-being. Accordingly, offering appropriate support improves employee engagement which often determines how much effort an employee is willing to commit to their jobs and the organization [16, 25, 46]. Indeed, organizational support is a key relational factor that improves employee engagement and motivate them to work for organizational success [47, 48].

Previous studies [47, 49] have also established the efficacy of perceived organizational support as a moderating variable that mitigates negative effects or strengthens positive relationships among various constructs or variables in a study. For instance, a study by [47] provided support for POS as a moderator of the link between servant leadership and psychological ownership. POS can be used to lessen the impact of work-related harassment on the target's intention to leave. [49] posited that POS can be used to strengthen employees' organizational citizenship behavior and performance relationship. According to [34], continual good treatment from the organization can increase employees' felt commitment to help the organization achieve its purpose. Thus, it is not surprising to see employees with strong perceived organizational support contributing more to job performance than those with poor perceived organizational support.

Based on the preceding review, it is hypothesized that:

*H₂*. *Perceived organization support has a moderating effect on the relationship between employee engagement and job performance.*

## Methodology

This section of the study focusses on the methodological issues underlying the study.

### The study organization, target population and sampling procedures

This study was conducted at the Cape Coast Teaching Hospital (CCTH) in the Central Region of Ghana. Although there are several hospitals in the Region, the Cape Coast Teaching Hospital, popularly known as "Interberton" is the largest and the major teaching hospital with a referral capacity [3]. Given its size and the role it play in the Region, the Cape Coast Teaching Hospital has diverse categories of nurses and patients based on specialty units. The hospital provides specialty care in ophthalmology, paediatrics, general surgery and internal medicine that makes it more attractive to patients from all parts of the country [3]. Notwithstanding the significant contribution of the hospital, its' performance has been affected in recent years. According to the Chief Executive Officer of the hospital, the decline in performance is partly due to the mpact of Covid– 19 Pandemic and inadequate support and employee engagement [3]. It is against this that this study is conducted in the Cape Coast Teaching Hospital.

The population for the study comprised nurses, medical doctors and midwives. The total population was 1,175. The stratified random sampling technique was used to select 291 respondents out of the total population. This sample size was based on [50] Sample Size Determination Table at a 95 percent confidence interval. The researcher used stratified random sampling technique because the respondents (nurses, medical doctors, and midwives) have different characteristics. Participants were selected based on their full-time employment status at the hospital and their willingness to participate in the study. Part-time staff, temporary workers, and those unwilling to participate were excluded. To ensure the sample accurately represents the population, each subgroup was sampled proportionately [51] as shown on Table 1. This involved listing all the nurses, doctors, and midwives and then randomly selecting participants from each list according to the calculated proportions. Prior to this, the table of random numbers was used to generate respondents (*see* Table 1). Given the total population of 1,175 medical staff, which included 711 nurses, 291 doctors, and 164 midwives, the stratified random sampling technique was employed to ensure proportionate representation of each subgroup in a final sample size of 292. The selection was done based on a 25% proportion to size. Again to reduce non response rate the sample size was increased to 320. This approach reduces selection bias and ensures that the sample reflects the diversity within the population.

### Data collection instruments, procedures and measurement of constructs

Data for this study was collected using the survey questionnaire. According to Esponda et al. (2016), the survey questionnaire provides confidentiality and privacy. The questionnaire was organized into four sections; A, B, C, and D. Section A gathered data on the demographic characteristics of respondents while section B focused on employee engagement. The 9-item

**Table 1. Sample size.**

| Employees | Population | Number sampled (0.25) |
|---|---|---|
| Nurses | 711 | 178 |
| Doctors | 291 | 73 |
| Midwives | 164 | 41 |
| | **1175** | **292** |

Source: CCTH (2021)

ISA engagement scale developed by [20] was used for assessing employee engagement. The scale composed of 3-item subscales; intellectual engagement, social engagement, and affective engagement. Academic research has found the ISA scale to be statistically valid and reliable. The Cronbach's Alpha in past studies was .897, .817 and .896 for intellectual engagement, social engagement, and affective engagement respectively. Section C of the questionnaire gathered data on perceived organizational support. The 8-item scale developed by [52] were used for assessing respondents perceived organizational support. Section D focused on employee job performance. In all, 15 items were adapted from the Individual Work Performance Questionnaire by [53].

The instrument consisted of three job performance dimensions: task performance, contextual performance, and adaptive performance. It has been reported that the Cronbach's Alpha in past studies ranges between .75 to .92. All items in the survey questionnaire, except those in section A were measured on a Five-Point Likert Scale ranging from *'1'- 'Strongly Disagree' to '5'- 'Strongly Agree'*. Out of the total 320 questionnaires issued, the researcher retrieved 310 successfully completed questionnaires, giving a return rate of 96.88%. The addition of 29 questionnaires to the sample size was prompted by previous studies who had problems in retrieving all administered questionnaires to meet their minimum sample sizes. According to [50], to meet the minimum sample size, a researcher can add as much as 10% of the sample size to the questionnaires being administered.

### Data collection procedure

Once the participants were selected, comprehensive training was provided to research assistants to standardize the data collection process. This training focused on ensuring that the administration of the survey was consistent across all respondents, with uniform instructions and clear guidelines for providing clarifications. The survey questionnaires, organized into sections on demographics, employee engagement, perceived organizational support, and job performance, were then self-administered to the participants between February 1st and March 28th, 2023. To encourage honest and unbiased responses, anonymity and confidentiality were emphasized, assuring participants that their responses would be kept confidential and used solely for research purposes. Throughout the data collection period, response rates were closely monitored to detect any non-response bias. Follow-up procedures were implemented to ensure balanced representation across all subgroups, with adjustments made to the sampling process if any group appeared underrepresented.

### Data processing and analysis

The responses were analysed and processed utilising the Statistical Package for Social Sciences (SPSS) version 3.3 software. The data at hand was subjected to analysis using the Partial Least Squares-Structural Equation Modelling 3.0 (PLS-SEM). The Partial Least Squares Structural Equation Modelling (PLS-SEM) methodology is employed to assess the relationships among the pathways within a theoretical model to minimise the residual variance associated with the endogenous components. The Partial Least Squares Structural Equation Modelling (PLS-SEM) technique is widely used by researchers to assess measurement models and evaluate structural models. It provides a comprehensive framework for analysing complex relationships between variables and allows for a thorough examination of the underlying constructs and their interrelationships. As stated by [54], the evaluation of a reflective PLS model's measurement model is a technique employed to determine the statistical reliability of the measurement model before conducting a structural analysis.

According to [54], to achieve statistical confidence, the measurement model must exhibit satisfactory levels of internal consistency reliability, convergent validity, and discriminant validity. The present study employed a systematic approach by utilising Smart-PLS, a widely recognised software tool in the field of research methodology. The assessment of internal consistency was conducted using Cronbach's alpha, rho A, and composite reliability measures. To assess convergent validity, the study employed factor loadings (also referred to as indicators) and the Average Variance Extracted (AVE) as recommended by [54]. As stated by [54], it is recommended that the values for indicator loading, Cronbach Alpha, and Composite reliability should be equal to or greater than 0.7 ($\geq 0.7$). As stated by [55], the Fornell-Larcker criterion, cross-loadings, and Heterotrait-Monotrait Ratio criterion are recognised statistical measures employed to assess the discriminant validity of a construct. For this study, we have chosen to utilise the Heterotrait-Monotrait Ratio criterion to evaluate the discriminant validity of the constructs under investigation. This criterion is particularly useful as it provides an upper limit for assessing discriminant validity, as demonstrated by previous research conducted by [55] and [54]. The HTMT (Heterotrait-Monotrait Ratio of Correlations) represents a highly reliable method for evaluating discriminant validity, surpassing both the Fornell-Larcker criterion and cross-loadings criterion in terms of effectiveness.

According to research, it is recommended that the Average Variance Extracted (AVE) should have a value that exceeds 0.5 to be considered acceptable. About the structural model, it has been established that R-square values of 0.25, 0.5, and 0.7 are considered weak, moderate, and substantial, respectively. The coefficient of determination (R2), which measures the proportion of variance in the endogenous components explained by the model, is widely considered as the primary indicator for assessing the predictive power of Partial Least Squares Structural Equation Modelling (PLS-SEM). The classification of values as weak, moderate, and substantial is based on the work of [54], where values of 0.25, 0.5, and 0.75 are assigned to these respective categories. The subsequent stage in establishing the structural model involved the computation of the regression coefficients between the validated latent variables.

According to [55] it is necessary to assess statistical significance by considering regression coefficients that are equal to or lower than the threshold of 0.05. According to [55] it is worth noting that a significant level up to 5% is commonly regarded as normal. The evaluation of the structural model was concluded by assessing the model's capability to forecast its F2 and Q2. The Q2 statistic, as proposed by Stone in 1974, is commonly employed to assess the predictive performance of a given model. Based on the given information, it can be observed that the exogenous construct exhibits varying levels of predictive relevance for a specific endogenous construct. Specifically, the predictive relevance values of 0.35, 0.15, and 0.02 indicate that the exogenous construct has a large, moderate, and small predictive relevance, respectively, for the endogenous construct in question. As stated by [56], F2 values of 0.02, 0.15, and 0.35 are indicative of small, medium, and large effects of an exogenous latent variable. According to [56] more realistic standards for small, medium, and large effect sizes are proposed to be 0.005, 0.010, and 0.025, respectively.

## Measurement model specification

The measurement model shows the indicators used to assess each construct. Thirty-two indicators were used to assess the three constructs in this model: employee engagement, perceived organizational support, and job performance. To measure employee engagement, 9 indicators were used. The adapted indicators from this construct were itemized as *IEE1*, *IEE 2*, *IEE3*, *SEE4*, *SEE5*, *SEE6*, *AEE7*, *AEE8 and AEE9*. To measure perceived organizational support, 8 indicators were used. The 8 indicators were itemized as *POS1*, *POS2*, *POS3*, *POS4*, *POS5*,

**Table 2. Construct reliability and validity.**

|  | Cronbach's Alpha | rho_A | Composite Reliability | (AVE) |
|---|---|---|---|---|
| POS | 0.876 | 0.875 | 0.910 | 0.669 |
| Job performance | 0.932 | 0.932 | 0.942 | 0.621 |
| Moderating Effect 1 | 1.000 | 1.000 | 1.000 | 1.000 |
| Employee engagement | 0.904 | 0.906 | 0.929 | 0.723 |

*POS6*, *POS7* and *POS8*. Finally, the 15-items scale that measured job performance was operationalized into *JOBP1*, *JOBP2*, *JOBP3*, *JOBP4*, *JOBP5*, *JOBP6*, *JOBP7*, *JOBP8*, *JOBP9*, *JOBP10*, *JOBSP11*, *JOBP12*, *JOBP13*, *JOBP14*, and *JOBP15*.

The model was specified and assessed in a reflective manner, adhering to established procedures commonly used for evaluating reflective models. The measurement model encompasses an evaluation of the reliability and validity of the scales and data utilised in the study. The evaluation of the reflective outer model entails the analysis of various aspects. These include assessing the reliabilities of the individual items, also known as indicator reliability. Additionally, the reliability of each latent variable is examined, along with measures of internal consistency such as Cronbach alpha, composite reliability, and rho_A. Construct validity is also considered, specifically convergent validity which is determined by the average variance extracted. Furthermore, discriminant validity is assessed using the HTMT ratio [57]. The measurement of construct reliability was conducted using two commonly employed methods: Cronbach's alpha (CA) and rho_A. Additionally, the reliability of the indicators was evaluated by examining the item loadings. Finally, the assessment of convergent validity was accomplished by calculating the average variance extracted (AVE). Discriminant validity was evaluated using the heterotrait-monotrait (HTMT) ratio, which is a commonly employed method in research. Table 2 presents the evaluation criteria utilised for the model.

As in Table 2, the Cronbach Alpha (CA) values for all the items exceeded the minimum threshold 0.7 [58]. These results indicate that the model has internal consistency. Furthermore, Table 2 revealed that the Composite Reliability (CR) for all the constructs were reliable [57]. This is because the constructs had CR scores higher than the 0.7 threshold [58]. Again, the rho A results for all the constructs were reliable because they all met the 0.7 minimum criteria. Finally, it was found that AVEs for the constructs accurately measured convergent validity because AVE attained values higher than 0.5.

## Discriminant validity

Discriminant Validity (DV) is frequently determined by the heterotrait-monotrait (HTMT) ratio because of its robustness and dependability [58]. The heterotrait-monotrait (HTMT) results for this study are presented in Table 3.

The results in Table 3 showed that all the constructs accurately measured discriminant validity because all the values are below the 0.9 threshold [59, 60].

**Table 3. Heterotrait-montrait ratio.**

| Construct | Employee Engagement | Moderating Effect 1 | Perceived support (POS) | Job performance |
|---|---|---|---|---|
| Engagement |  |  |  |  |
| Moderating Effect | 0.668 |  |  |  |
| POS | 0.818 | 0.641 |  |  |
| Job performance | 0.820 | 0.607 | 0.839 |  |

**Table 4. Inner VIF values.**

|  | Job Performance |
| --- | --- |
| Perceived Organizational Support | 2.322 |
| Moderating Effect 1 | 1.802 |
| Employee engagement | 2.494 |

## *Collinearity s*tatistics (VIF)

The results from the collinearity test are presented in Table 4. For the VIF to be acceptable, it must meet a threshold less than 5 [61]. The Variance Inflation Factor (VIF) was used to measure collinearity in this study.

Table 4 presented the VIF scores for the inner model of the constructs used in the model. Results from Table 4 showed that all the VIF scores are less than 5. The VIF scores for the inner model therefore, showed that there is no threat of common method bias for all the constructs.

## Outer loadings

The results in Table 5 showed that the factor loading for all the indicators had loadings of more than the 0.7 threshold except IEE1, IEE2, AEE7, AEE9, POS2, POS6, POS7, JOBP4, JOBP8, JOBP9, JOBP13, JOBP14, and JOBP15. These indicators, as they loaded below 0.7 were eliminated to improve the constructs reliabilities. This indicates that all the factors strongly

**Table 5. Indicator loadings.**

|  | Loadings | T Statistics | P Values |
| --- | --- | --- | --- |
| IEE3 | 0.852 | 24.533 | 0.000 |
| SEE4 | 0.840 | 28.699 | 0.000 |
| SEE5 | 0.878 | 41.503 | 0.000 |
| SEE6 | 0.870 | 35.584 | 0.000 |
| AEE8 | 0.811 | 21.417 | 0.000 |
| JOBP1 | 0.844 | 49.514 | 0.000 |
| JOBP10 | 0.743 | 28.865 | 0.000 |
| JOBP11 | 0.790 | 27.632 | 0.000 |
| JOBP12 | 0.769 | 30.063 | 0.000 |
| JOBP2 | 0.807 | 38.774 | 0.000 |
| JOBP3 | 0.832 | 39.674 | 0.000 |
| JOBP5 | 0.825 | 31.622 | 0.000 |
| JOBP6 | 0.747 | 21.757 | 0.000 |
| JOBP7 | 0.793 | 24.665 | 0.000 |
| JOBP9 | 0.719 | 17.361 | 0.000 |
| POS * Engagement | 1.781 | 84.256 | 0.000 |
| POS1 | 0.849 | 34.747 | 0.000 |
| POS3 | 0.854 | 30.399 | 0.000 |
| POS4 | 0.819 | 32.871 | 0.000 |
| POS5 | 0.734 | 29.807 | 0.000 |
| POS8 | 0.828 | 27.872 | 0.000 |

P > 0.05

measure the constructs they purport to measure especially as seen with their respective p values. The outer loadings were all statistically significant because they had p values lesser than 0.05 (p < 0.05). Thus, in all instances, the T-statistics for the indicators were larger than 1.96.

## Structural model specification

The structural model has one exogenous and two endogenous constructs of which one was a moderating variable. Employee engagement was the exogenous latent variable while perceived organizational support was used as an endogenous moderating variable. The second endogenous construct was job performance. The structural model went through the following assessment. The standardised path weights, also known as measurement loadings, were used to quantify the strength of the connections between the components and the indicator variables. These loadings range from 0 to 1, indicating the extent to which the indicator variables are influenced by the underlying components. According to research findings, it has been observed that there is a positive relationship between the magnitude of loadings and the robustness and reliability of the measurement model. In other words, as the loadings increase in size, the measurement model tends to exhibit greater robustness and reliability. This suggests that higher loadings are indicative of a stronger and more accurate measurement model.

According to [55], the measurement loadings in a reflective model were generally found to be above 0.70. The assessment of the contributions made by both direct and indirect predictors to the variance observed in the dependent variable was conducted through the utilisation of path coefficients, specifically unstandardized beta coefficients. In order to assess the extent to which the predictors influenced the variations in the dependent variable, the investigators employed the Effect size ($f^2$) metric, as suggested by [57]. According to [55], values greater than 0.35 are considered strong, values between 0.15 and 0.35 are classified as moderate, and values between 0.02 and 0.15 are categorised as weak. The assessment of this was conducted using the R-square, which is widely recognised as the predominant effect size measure in path models. It has been suggested by [57] that in order to achieve this objective, it is advisable to use tentative cut-off points. The results with a value greater than 0.67 are characterised as "substantial," while those with a value between 0.33 and 0.67 are considered "moderate," and those with a value between 0.19 and 0.33 are deemed "weak." The study also examined the Adjusted R-squared and coefficient of determination. The utilisation of Adjusted R-squared was employed to elucidate the manner in which alterations in one variable exert an influence on the other. The PLS-SEM algorithm and the bootstrap procedure were employed to analyse the direct effect, total indirect effect, specific indirect effects, and total effect. The outcomes presented in the results section provide the necessary data for the authors to perform a moderation analysis, as recommended by [54]. Moreover, the obtained results facilitated the examination of both single and multiple mediation models, specifically parallel and serial mediation. Ultimately, the $Q^2$ was subjected to analysis utilising the PLS predictor. The Q-square statistic is a measure of predictive relevance, indicating the extent to which a model demonstrates predictive relevance. A Q-square value greater than zero is generally considered desirable, indicating that the model has predictive relevance. Moreover, $Q^2$ provides evidence for the predictive significance of the endogenous constructs. Q-square values greater than zero are indicative of well-reconstructed values and suggest that the model possesses predictive relevance. Put simply, a $Q^2$ value greater than 0 indicates that the model possesses predictive relevance, while a value below 0 suggests the opposite.

The results in Table 6 indicated that perceived organizational support moderated the relationship between employee engagement and job performance of medical staff at Cape Coast Teaching Hospital given its interaction effect (Beta = 0.034; t = 2.416; p = 0.002: p < 0.05).

**Table 6.  Path co-efficient, effect size and predictive relevance.**

|  | Beta | F-Squared | T Statistics | P Values |
|---|---|---|---|---|
| POS -> Job performance | 0.609 | 1.381 | 17.678 | 0.000 |
| Moderating Effect -> Job performance | 0.034 | 0.017 | 2.416 | 0.002 |
| Engagement -> Job performance | 0.440 | 0.669 | 10.767 | 0.001 |

** <0.05

Hence, given a high score on perceived organizational support, employee engagement increases its importance in explaining job performance among medical staff at the hospital. The Effect size of the moderator showed a medium statistically significant positive variance in the job performance of medical staff (f-squared = 0.017) as stipulated by [56]. Further, the results indicated that perceived organizational support made a statistically significant positive contribution to causing a positive variation in the job performance of medical staff at Cape Coast teaching hospital (Beta = 0.609; t = 17.678; p = 0.000: p<0.005). Thus, it can be expressed that a unit increase in perceived organizational support causes 0.609 significant improvements in the job performance of the medical staff and vice versa. The Effect size score further shows that perceived organizational support caused a large significant positive variance in the job performance of medical staff (f-squared = 1.381).

Similarly, employee engagement made a statistically significant positive contribution to causing positive variation in job performance of medical staff at Cape Coast teaching hospital (Beta = 0.440; t = 10.767; p = 0.001: p < 0.05). Thus, a unit increase in employee engagement causes 0.440 significant improvements in the job performance of medical staff at Cape Coast teaching hospital. The Effect size score further showed that employee engagement caused a large statistically significant positive variance in the well-being of employees (f squared = 0.669).

## Coefficient of determination

The coefficient of determination ($R^2$) vary from 0 to 1, with greater values indicating better effects of prediction ability (Hair et al., 2016). $R^2$ values of 0.75, 0.50, and 0.25, for example, are considered high, moderate, and low (Hair et al., 2019). The Effect size ($f^2$), which indicates the variation in $R^2$ when a certain predictive construct is excluded from the framework, was utilized to determine if the eliminated construct has a substantial impact on the endogenous variables [54]. According to [55], Effect size ($f^2$) scores of 0.02, 0.15, and 0.35 imply low, moderate, and high impacts respectively. The coefficient of determination ($R^2$) and Effect size ($f^2$) results for the study are presented in Table 7.

The results in Table 7 showed that employee engagement together with perceived organizational support and its moderating effect accounted for a strong variance in job performance (R-squared = 0.884) when all other factors not captured in this study but are affecting the job performance of medical staff at Cape Coast teaching hospital are statistically controlled for.

**Table 7.  Coefficient of determination.**

|  | R Square | R Square Adjusted |
|---|---|---|
| Job performance | 0.884 | 0.883 |

$R^2$ > 10%

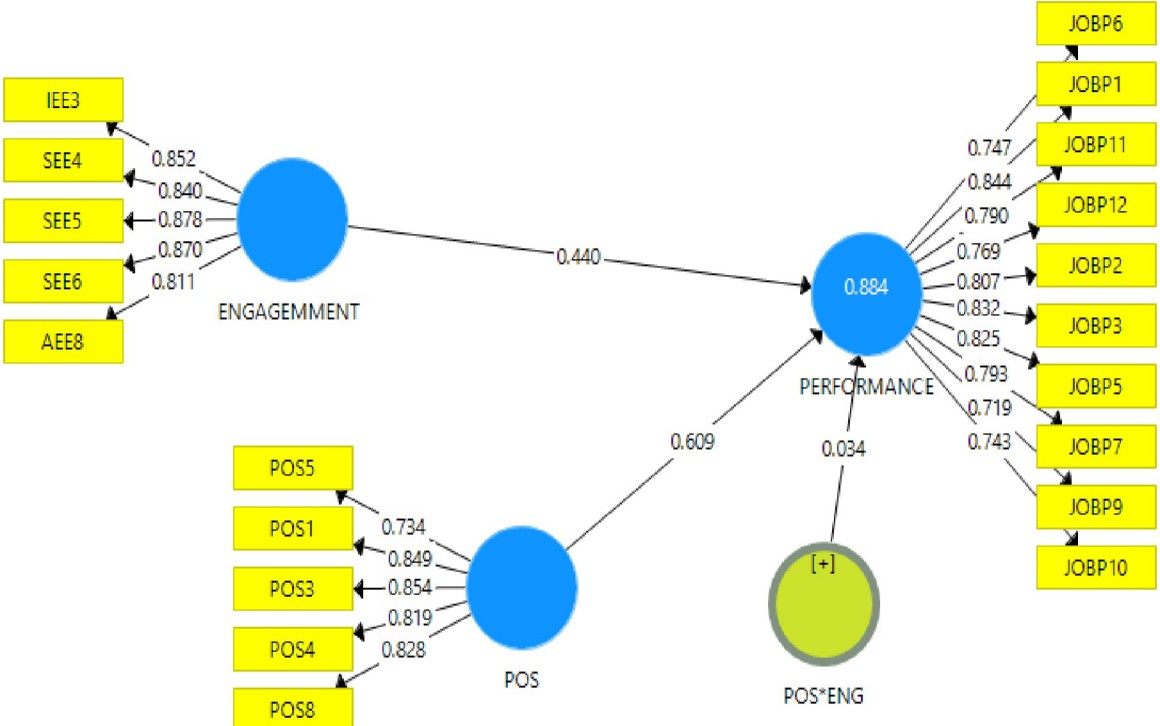

**Fig 2. Structural model showing the moderating effect of perceived organizational support on the link between employee engagement and job performance of medical staff at Cape Coast Teaching Hospital.**

Thus, 88.4% variance in job performance of medical staff can be attributed to changes in employee engagement, perceived organizational support and its interaction effect. The structural model is presented pictorially in Fig 2.

The positioning of Perceived Organizational Support in the upper-right quadrant of the Importance-Performance Map Analysis (IPMA) as shown in Fig 3 highlights its significant impact on both employee engagement and job performance. High Perceived Organizational Support suggests that employees feel valued and supported by the organization, which directly enhances their engagement. When employees perceive strong organizational support, they are more likely to be committed, motivated, and emotionally invested in their work. This increased engagement leads to better job performance, as engaged employees are more productive, innovative, and willing to go the extra mile. Conversely, the lower placement of Job Performance in the diagram indicates that while it is currently less prioritized, its improvement could further enhance overall performance. Employee Engagement, centrally located, reflects moderate importance and performance, suggesting room for growth. By focusing on maintaining and improving Perceived Organizational Support, the organization can foster higher employee engagement, which in turn, can lead to improved job performance and overall organizational success.

The simple slope analysis presented in Fig 4 graphically illustrates the relationship between employee engagement and job performance, moderated by perceived organizational support. The slope analysis in Fig 4 prove that while increased employee engagement generally leads to better job performance, the effect is significantly amplified when organizational support is high. The implications are clear: hospital management should prioritize enhancing perceived organizational support to maximize the positive impact of employee engagement on job

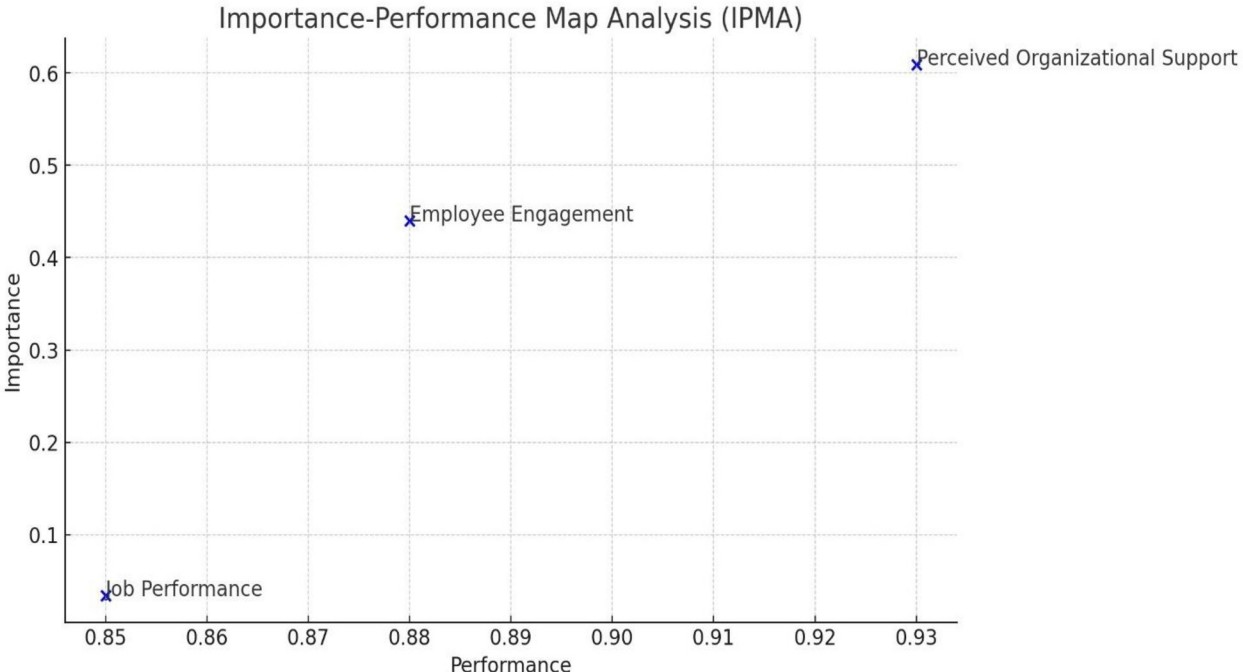

**Fig 3. Important performance map analysis.**

performance. This can be achieved through targeted support programs, fair treatment, resource provision, and career development opportunities, ensuring that employees feel valued and supported, ultimately leading to improved performance.

## Discussions

This study examined the moderating effect of perceived organizational support on the relationship between employee engagement and job performance of medical staff at the Cape Coast Teaching Hospital. The discussion of the results of this study is organized into two sections based on the two hypotheses formulated for the study.

**Hypothesis 1**: *Employee engagement has a significant effect on Job Performance of Medical Staff*

This hypothesis sought to evaluate the influence of employee engagement on the job performance of staff of the Cape Coast Teaching Hospital. The results indicate that employee engagement has significant positive influence on employee job performance (Beta = 0.440; t = 10.767; p = 0.001: p < 0.05) as shown in Table 5 and Fig 2. The effect of employee engagement on job performance was strong as the $f^2$ for this relationship ($f^2$ = 0.669) was above 0.35 [18]. The results imply that a change in employee engagement will have a positive effect on employees' job performance. As the p-value is less than 0.05, it is considered significant, meaning that employee engagement has a substantial beneficial influence on the job performance of staff of the Cape Coast Teaching Hospital. These results may be attributed to intrinsic and extrinsic factors influencing employee engagement at the hospital. It may also be attributed to fair treatment of staff by supervisors and management which usually improves staffs' intellectual, social and affective emotions to work. The perception of being treated fairly can improve employee engagement where they can voluntarily help coworkers who experience difficulties in terms of

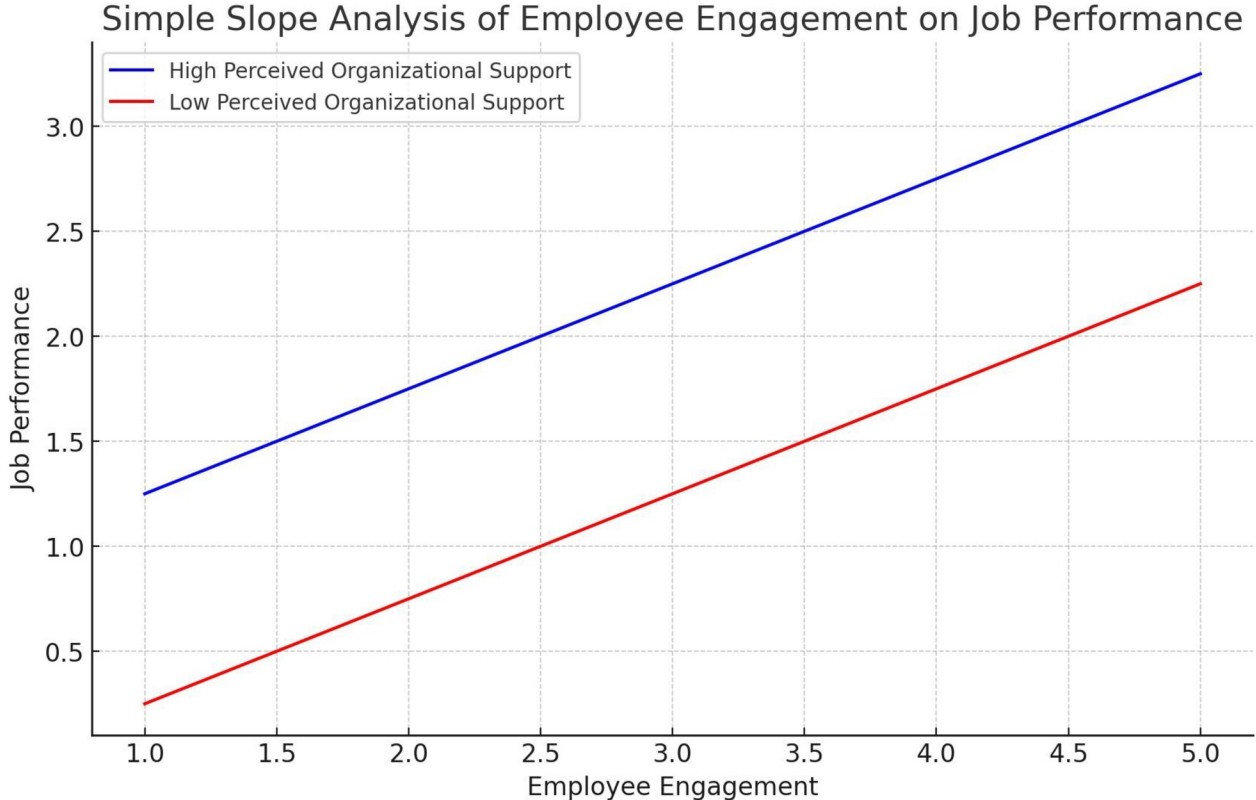

## Simple Slope Analysis of Employee Engagement on Job Performance

**Fig 4. Slope analysis.**

work, work hard to exceed the expectations of the Hospital, and participate fully in hospital activities outside their daily schedule [43]. Arguably, these conditions can propel superior performance among staff of the hospital.

The preceding findings are consistent with the findings of [41] that employee engagement has significant effect on the job performance of managerial employees from small-medium scale organizations in India. The findings are equally consistent with the submissions by [25] that drivers of employee engagement such as work-life balance, communication, and leadership had a significant effect on employee job performance. The results also support the recent empirical survey conducted by [45] which found that employee engagement has a significant and positive effect on the job performance of staff of the banking sector in Ethiopia. Furthermore, the results of this study corroborate the findings of [45] and [44] that employee engagement has a significant, positive effect on their job performance in Lebanon and Pakistan respectively. Given the positive association between employee engagement and job performance as found in this and previous studies [25, 41–45] it can be said that employee engagement is one of the most powerful determinants of employee job performance. This implies that management of Cape Coast Teaching Hospital can help improve the job performance of their staff by instituting HR policies such as effective leadership and fair treatment to improve their employees' engagement.

**Hypothesis 2**: *Perceived organizational support has a moderating effect on the link between employee engagement and job performance of medical staff*

The second hypothesis sought to assess the moderating effect of perceived organizational support on the link between employee engagement and job performance of medical staff at the Cape Coast Teaching Hospital. The results of this study, as presented in Table 5 and Fig 2 indicate that perceived organizational support is a moderator between employee engagement and job performance (Beta = 0.034; t = 2.416; p = 0.002: p<0.05). The moderating effect of perceived organizational support on the link between employee engagement and job performance was however, weak as the $f^2$ ($f^2$ = 0.017) was below 0.02 [57]. The results imply that an increase in perceived organizational support will improve the link between employee engagement and employees' job performance. As the p-value is less than 0.05, it is considered significant, meaning that perceived organizational support has a substantial beneficial influence on the link between employee engagement and job performance of staff at the Cape Coast Teaching Hospital. This means the effect of employee engagement on job performance of medical staff of the Cape Coast Teaching Hospital was affected by the presence of perceived organizational support. Put differently, in the context of a developing country, medical staff who receive adequate support are more likely to be good job performers. Based on the findings of study, the researcher accepted the hypothesis that perceived organizational support has a moderating effect on the nexus between employee engagement and job performance of staff at Cape Coast Teaching Hospital.

The preceding results imply that when there are adequate support systems from the hospital, medical staff are automatically motivated to increase on their levels of engagement. Giving adequate support to medical staff which includes fair distribution of resources, designing equitable compensation systems and schemes, providing career advancement opportunities and providing explanations in every decision taken, will improve employee engagement and their job performance. These analyses are informed by the social exchange theory and organizational support theory which holds that on the basis of social exchange, employees strive to repay the organization for a high level of support in the form of superior performance. Indeed, employees who perceive high organizational support may be absorbed emotionally, affectively, socially and cognitively to ensure that the organization achieve her stipulated goals and objectives. This supports the view by [34] that adequate support and good treatment of employees from their employers increases employees' felt commitment and dedication to the goals and objectives of the organization.

## Theoretical implication

The theoretical contribution of the present study lies in its expansion of existing knowledge and utilisation of the social exchange theory, as examined in the relevant literature. Through a comprehensive examination of the initial theory put forth by [13] and its subsequent elaboration by [14], this research aims to enhance the comprehension of readers regarding the theory's conceptual framework and its practical application within a specific empirical context in Ghana. This study has made a valuable contribution to the existing body of knowledge by examining the impact of perceived organisational support on the relationship between employee engagement and job performance. By investigating these variables and their interactions, this research sheds light on how they can collectively enhance the effectiveness of the human resource department. Again, this study adds not just a developing-country viewpoint, but also data from the health sector to the employee engagement literature which has previously been dominated by perspectives from developed countries and other sectors, including manufacturing, finance, and banking.

### Managerial implications

The findings of this study have potential implications for managerial practice. First, the results show that employee engagement has significant, positive influence on employee job performance. This means that management of the Cape Coast Teaching Hospital can promote superior employee performance by adopting HR practices such as, positive organizational politics, job fit and creation of a conducive psychological climate to improve employee engagement in the organization. In other words, management and the organization can profit from these findings by stimulating employees' work engagement through appropriate engagement-evoking policies. Finally, the results have also shown that perceived organizational support has significant influence on the link between employee engagement and job performance. This implies that management can strengthen the link between employee engagement and job performance only when their HR policies also help to improve perceived organizational support. In other words, where management fails to consider or take into account the effect of perceived organizational support on the link between employee engagement and job performance, the actual performance of staff can be stalled.

## Conclusion, limitations and suggestions for future research

This study investigated the relationship between employee engagement, perceived organizational support and job performance of medical staff at the Cape Coast Teaching Hospital. The main aim of the study was to determine the effect of perceived organizational support on the link between employee engagement and job performance of staff at the Cape Coast Teaching Hospital. The results revealed that perceived organizational support moderates this link. In line with previous studies [25, 41–45], we can conclude that employee engagement has significant, positive influence on employees' job performance. Much like other previous studies [47, 49], we can equally conclude that perceived organizational support moderates the link between employee engagement and job performance. The study's findings highlight the importance of implementing targeted support programs and continuous engagement initiatives. For instance, hospital management should develop and implement support programs tailored to the specific needs of nurses and midwives. These programs can include mentorship opportunities, professional development workshops, and access to mental health resources. Additionally, establishing ongoing initiatives to keep employees engaged, such as regular feedback sessions, team-building activities, and recognition programs for outstanding performance, is crucial. Also, promoting leadership practices that are inclusive, transparent, and supportive is also essential. Leaders should regularly communicate with staff, provide clear expectations, and offer support and resources needed to perform their duties effectively. Furthermore, ensuring that HR policies are fair and consistently applied, including equitable compensation, opportunities for career advancement, and fair distribution of workloads, is vital for enhancing job performance and employee satisfaction.

Notwithstanding the contribution of this study, there are some few limitations. Firstly, it was conducted exclusively at the Cape Coast Teaching Hospital, which may limit the generalizability of the findings to other regions or sectors. Finally, cultural differences between Ghana and other countries may limit the applicability of the findings to other contexts. Owing to these limitations, studies could be conducted to include more hospitals in different regions for a cross-regional analysis. Additionally, while the study focuses on perceived organizational support as a moderating variable, other potential moderating factors such as leadership style or work-life balance were not considered.

## Supporting information

**S1 File. Data for article.**
(CSV)

## Author Contributions

**Conceptualization:** Felix Kwame Opoku.

**Data curation:** Richard Kofi Boateng.

**Formal analysis:** Richard Kofi Boateng.

**Validation:** Richard Kofi Boateng.

**Writing – original draft:** Richard Kofi Boateng.

**Writing – review & editing:** Felix Kwame Opoku, Richard Kofi Boateng.

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
