## [Decision Letter · Decision Letter 0]

17 Jul 2024

PONE-D-24-19644Employee engagement, perceived organizational support, and job performance of medical staff at the Cape Coast Teaching HospitalPLOS ONE

Dear Dr. Boateng,

Thank you for submitting your manuscript to PLOS ONE. After careful consideration, we feel that it has merit but does not fully meet PLOS ONE’s publication criteria as it currently stands. Therefore, we invite you to submit a revised version of the manuscript that addresses the points raised during the review process.

We look forward to receiving your revised manuscript.

Kind regards,

Ayatulloh Michael Musyaffi

Academic Editor

PLOS ONE

Journal Requirements:

2. In the online submission form, you indicated that [The data underlying the results presented in the study are available from the corresponding author upon reasonable request.]. 

Reviewers' comments:

Reviewer's Responses to Questions

**Comments to the Author**

1. Is the manuscript technically sound, and do the data support the conclusions?

Reviewer #1: Yes

Reviewer #2: Yes

2. Has the statistical analysis been performed appropriately and rigorously? 

Reviewer #1: Yes

Reviewer #2: Yes

3. Have the authors made all data underlying the findings in their manuscript fully available?

Reviewer #1: Yes

Reviewer #2: Yes

4. Is the manuscript presented in an intelligible fashion and written in standard English?

Reviewer #1: Yes

Reviewer #2: Yes

5. Review Comments to the Author

Reviewer #1: 1. This research is very simple, you have shown the position of this research compared to previous research. My question is, what is the urgency of this research...?

2. Some of the references used still use old years.

3 The author must explain why you chose the Cape Coast Teaching Hospital (CCTH) as the research object? in the methodology section

4. You need to explain what the shortcomings of this research are to provide material for further research development

Reviewer #2: This study examined the moderating effect of perceived organizational support on the link between employee engagement and the job performance of medical staff at the Cape Coast Teaching Hospital. Data was collected by using an explanatory research design and a quantitative approach. The results revealed that employee engagement significantly influences job performance and that perceived organizational support moderates this relationship. These results are relevant to confirm the research hypotheses.

I consider that this paper should attend to the following recommendations for improvement:

1. In the introduction section, I think it would be a good idea to elaborate on the practical implications of the research. Including a brief discussion on how the findings might benefit hospital management and policy-making in the healthcare sector. Also, adding recent statistics on employee engagement and job performance in healthcare could provide better insights on the relevance of this research.

2. In the literature review, it is very important to cover relevant theories and previous studies. To improve this section, consider adding a comparative analysis of similar studies conducted in different cultural or regional contexts. This would highlight the uniqueness and relevance of the current study. I suggest adding a conceptual mal or diagram to summarize key findings of the existing literature as well as an author's table where you explain the most relevant research on this topic.

3. The methodology is robust and well-detailed, but it could benefit from a more in-depth discussion on the selection criteria for participants and potential biases. Adding a section on how the data collection process was standardized across different respondents would improve transparency. I also suggest to include an explanatory diagram explaining each step of your research.

4. The results section is clearly presented and supported by appropriate statistical analysis. However, to enhance the depth of the analysis, consider incorporating Importance-Performance Map Analysis (IPMA) and Multi-Group Analysis (MGA) as part of the sensitivity analysis. These additions would provide the best understanding of the data and highlight key areas for improvement or further investigation. Adding visual aids like graphs and charts also can explain findings and recommendations in a better way.

5. The conclusions are well-supported by the data, but the section could be strengthened by including more specific recommendations for hospital management and contrasting findings with previous studies. Mention also, how each of the objectives of this research was completed, it could include a table or a diagram to explain this.

6. The references section is comprehensive and follows a standard format. Authors should review the citation style, due to they initiate several paragraphs with a reference indication at the beginning of them. The paper needs to have an in-depth revision of grammatical review.

6. PLOS authors have the option to publish the peer review history of their article (what does this mean?). If published, this will include your full peer review and any attached files.

Reviewer #1: **Yes: **Christian Wiradendi Wolor

Reviewer #2: **Yes: **Eduardo Ahumada-Tello

---

## [Author Response · Author response to Decision Letter 0]

29 Aug 2024

Data for article has been added as supporting information.

---

## [Decision Letter · Decision Letter 1]

26 Nov 2024

Employee engagement, perceived organizational support, and job performance of medical staff at the Cape Coast Teaching Hospital

PONE-D-24-19644R1

Dear Dr. Boateng,

We’re pleased to inform you that your manuscript has been judged scientifically suitable for publication and will be formally accepted for publication once it meets all outstanding technical requirements.

Kind regards,

Mohsin Shahzad

Academic Editor

PLOS ONE

Additional Editor Comments (optional):

Reviewers' comments:

Reviewer's Responses to Questions

**Comments to the Author**

1. If the authors have adequately addressed your comments raised in a previous round of review and you feel that this manuscript is now acceptable for publication, you may indicate that here to bypass the “Comments to the Author” section, enter your conflict of interest statement in the “Confidential to Editor” section, and submit your "Accept" recommendation.

Reviewer #1: All comments have been addressed

2. Is the manuscript technically sound, and do the data support the conclusions?

Reviewer #1: Yes

3. Has the statistical analysis been performed appropriately and rigorously? 

Reviewer #1: Yes

4. Have the authors made all data underlying the findings in their manuscript fully available?

Reviewer #1: Yes

5. Is the manuscript presented in an intelligible fashion and written in standard English?

Reviewer #1: Yes

6. Review Comments to the Author

Reviewer #1: Several notes in the previous article have been corrected by the author, especially regarding the novelty, uniqueness and significance of this research compared to previous research.

7. PLOS authors have the option to publish the peer review history of their article (what does this mean?). If published, this will include your full peer review and any attached files.

Reviewer #1: **Yes: **Christian Wiradendi Wolor

---

## [Editor Report · Acceptance letter]

6 Dec 2024

PONE-D-24-19644R1 

PLOS ONE

Dear Dr. Boateng, 

I'm pleased to inform you that your manuscript has been deemed suitable for publication in PLOS ONE. Congratulations! Your manuscript is now being handed over to our production team.

Kind regards, 

on behalf of

Dr. Mohsin Shahzad 

Academic Editor

PLOS ONE